# Acceptability of Digital Adherence Technologies to support people with drug-susceptible TB in South Africa

Tanyaradzwa Nicolette Dube[1]*, Fezeka Mboniswa[1], Liza de Groot[2],
Siyanda Khumalo[1], Noriah Maraba[1], Nontobeko Ndlovu[1], Nontobeko Mokone[1],
Lebohang Masia[1], Katherine Fielding[3,4], Degu Jerene[2], Zewdneh Shewamene[3],
Bianca Gonçalves Tasca[2], Adrian Leung[2], Salome Charalambous[1,4]

1 The Aurum Institute, Johannesburg South Africa, 2 KNCV Tuberculosis Foundation, The Hague,
Netherlands, 3 London School of Hygiene & Tropical Medicine, London, United Kingdom, 4 School of
Public Health, University of Witwatersrand, Johannesburg, South Africa

* tndube@auruminstitute.org

## Abstract

### Background

Tuberculosis (TB) remains a challenge in South Africa, with an estimated 280,000 new cases reported in 2022. Digital Adherence Technologies (DATs) may be important tools to improve adherence to TB treatment. However, there is limited knowledge about facilitators and barriers to implementing these technologies in South Africa.

### Methods

This qualitative study was embedded within the Adherence Support Coalition to End TB cluster-randomised trial which implemented the two technologies: a smart pillbox and/or a cell phone-based strategy similar to 99DOTS (labels), plus differentiated care, to support people with drug-sensitive TB (PWTB). In-depth interviews were conducted with purposively sampled participants comprising PWTB, Healthcare Workers (HCWs) and stakeholders. Inductive thematic analysis was used for data analysis, while the Unified Theory of Acceptance and Use of Technology (UTAUT) framework served as the overarching framework to synthesize and summarise the findings.

### Findings

Sixty-eight interviews were conducted: 35 with PWTB and 33 with HCWs and stakeholders. Facilitating factors for the pillbox were alarm reminder, storage, ease of use, not requiring a phone, social support, and portability. Barriers to the pillbox were that some found the box unportable and others experienced box malfunctioning. Facilitating factors for the labels were ease of use among the young, daily confirmation messages, social support, privacy, and portability. Barriers to implementing these DATs

**Data availability statement:** All relevant data are within the paper and its Supporting information files.

**Funding:** This study was supported by Unitaid in the form of a grant awarded to KNCV (Grant # 2019–33-ASCENT) in the form of a salary for TND, FM, LG, SK, NN, NM, LM, KF, DJ, ZS, BGT, AL and SC. The specific roles of this author are articulated in the 'author contributions' section. The funders had no role in study design, data collection and analysis, decision to publish, or preparation of the manuscript.

**Competing interests:** The authors have declared that no competing interests exist.

were older age, illiteracy, forgetting to send the SMS, lacking understanding, no cell phone access, power cuts, and unresolvable technical glitches. While differentiated care improved the client-provider relationship, some PWTB felt home visits were stigmatising. HCWs and stakeholders expressed willingness to scale up DATs because they improved their working conditions, and it was advantageous for the PWTB.

## Conclusions

Smart pillbox and labels had features which favoured their acceptability and there were also DAT specific barriers. DATs may improve person-centredness in TB care. Future guidelines should consider acceptability, differing situations, and allowing flexibility to possibly increase uptake and utilisation of DATs.

## Introduction

In 2022, 7.5 million people were affected with Tuberculosis (TB) globally, and efforts to reduce TB incidence are not enough to reach global targets [1]. TB incidence in South Africa was 280,000 in 2022 [1].Poor adherence to treatment can lead to poor end of treatment outcomes, developing TB drug resistance and further TB transmissions [2]. There are a number of factors associated with poor-adherence, which include, forgetting to take treatment; poor relationship with healthcare workers (HCWs); feeling better before completing treatment; fear of stigma; side effects; multi-morbidities; lack of food; substance abuse; and mental health illness [3,4]. Directly observed therapy (DOT) has been used to support people with TB (PWTB) to improve adherence [5]. DOT is a monitoring strategy which can be facility-based, whereby patients visit a health facility daily to have a HCW observe their medication intake; or community-based, where a HCW or trained treatment supporter supervises the patient's treatment at home [5]. However, despite offering intensive TB monitoring, DOT strains both PWTB and HCWs. For example, PWTB may have to frequently walk long distances to the health facility or incur travel costs and conducting community-based DOT strains financial and human resources on the health system [6,7]. For this reason, DOT implementation is limited in South Africa with challenges such as shortage of staff, difficulty for PWTB to access the facility as well as perceptions of low autonomy and lack of confidentiality having been reported [7]. Given the challenges of DOT, Digital Adherence Technologies (DATs) such as the smart pill box and video support treatment have been recommended by the World Health Organisation and the South African National Strategic Plan 2023–2028 [8,9]

DATs may be important tools to improve adherence to TB treatment. Cluster randomized trials conducted in South Africa and China found improvement in adherence among PWTB who used the smart pill box [10,11]. While interest in DATs is growing, there is limited knowledge about facilitators and barriers to implementing these technologies in low- and middle-income countries such as South Africa [9]. A few studies assessed the acceptability and feasibility of DATs among PWTB, HCWs and stakeholders in low and middle-income countries [12–14]. Only two qualitative

studies have previously evaluated DATs in South Africa, but they exclusively focused on the smart pillbox [12,13]. One of the studies was on people with drug-resistant TB only and the other on HCWs and stakeholders only. The acceptability of a cell phone-based strategy known as 99DOTS (labels) is unknown in South Africa. To address this gap, we investigated key factors influencing the acceptability and feasibility of smart pill boxes, and labels requiring shorts message service (SMS), with differentiated care among adults with drug-sensitive TB, HCWs, and stakeholders in South Africa.

## Methodology

### Parent study

This qualitative study is nested in the Adherence Support Coalition to End TB (ASCENT) cluster-randomized trial (CRT), funded by Unitaid, which implemented two DATs in South Africa, Tanzania, the Philippines, and Ethiopia and one DAT in Ukraine. The two DATs were the smart pillbox and medication labels with SMS (similar to 99DOTS). Differentiated care was offered based on number of missed doses. The South Africa trial evaluated the effectiveness of DATs by comparing treatment outcomes between intervention versus standard of care clusters in Tshwane (36 clinics) and Bojanala (24 clinics) districts, Gauteng and North-West provinces, respectively.

### DAT intervention and differentiated care

We evaluated the acceptability of two DAT interventions and differentiated care in South Africa. The first DAT intervention was the smart pillbox which stores medication, has a daily alarm reminder which beeps when it is time to take medication, and digitally records box-opening events onto an adherence platform (Everwell). In addition, a yellow light illuminated to remind participants to go for their refill visits. It worked in real-time (online) mode and offline in settings with low connectivity. The second DAT were the labels pasted on the medication blister pack, which displayed a numeric code. Participants were expected to use their own cell phones to send the code via an SMS to a toll-free number every day after taking medication. The SMS was digitally recorded onto the platform, after which the participant received a confirmation SMS. An SMS reminder would be sent in the evening at 6PM if the box was not opened or an SMS was not sent by the participant using the label. HCWs used the adherence platform to remotely monitor patients in intervention facilities in real time for both DATs. A box non-opening or SMS not sent by a participant in the labels arm was considered a proxy for missed dose. Differentiated care was given based on missed doses and consisted of a reminder SMS for one missed dose, a phone call for two or three missed doses, and a home visit, plus motivational counselling for four or more missed doses [15].

### Study setting and population

In-depth interviews (IDIs) were conducted at public health clinics where care for TB was offered in Bojanala and Tshwane districts in South Africa. Interviews with stakeholders were conducted at district and provincial offices.

### Data collection

We conducted interviews with PWTB who were on TB treatment from 15 facilities representing the two districts. We purposively sampled participants to represent: (i) those who have documented good adherence; (ii) those with poorer adherence for whom differentiated response to patient adherence had been initiated; (iii) those who opted out of using DAT (but were still on TB treatment at the health facility). Good adherence was defined as 90% or more adherence and bad adherence was less than 90% adherence. In addition, we conducted interviews with HCWs from 15 implementing sites, stakeholders such as district and provincial coordinators, treatment supporters and research interns who were supporting the TB nurses with implementation. The interns obtained informed consent from the participants, enrolled the participants in the trial, followed up on those who missed doses and reported technical problems with the DAT. We also ensured diversity of the clinical role, facility location (rural vs urban), and type of DAT implemented.

We piloted the interview guide for PWTB in March 2022 and for HCWs and stakeholders in June 2022. PWTB fitting into the different categories mentioned above were identified on the adherence platform. Potential participants were all approached by telephone or email as in the case of HCWs and stakeholders. The interviews were conducted in-person by male and female trained researchers with either diplomas or honours degrees (FM) or Masters degree (TD). We took measures to ensure that research interns engaged in the implementation process did not conduct interviews in clinics where they were providing assistance, so that there was no pre-existing relationship between patient participants and the moderator, except for rapport building during the informed consent process. However, in the case of the HCWs and stakeholders, there were pre-existing relationships with the research team, as many had attended trainings that were coordinated by the research team. To manage the relationship, the interviewers encouraged the HCWs to feel free to share their honest opinions and assured them that their responses were confidential. During interviews, the interviewers emphasized neutrality and created a respectful, non-judgmental space to mitigate social desirability bias and support candid engagement [16].

Each interview was audio recorded to ensure the capture of all points, and in addition, a notetaker documented key information. The interviews ranged from 25 minutes to 78 minutes for PWTB interviews and 25 minutes to 130 minutes for HCWs and stakeholders. No one else was present in the interviews besides the participants and researchers. Saturation during data collection was achieved through regular debriefing discussions with the data collectors and interviews were stopped when no new information emerged [17,18]. There were no repeat interviews done. Participants were compensated for their time with ZAR 150 (equivalent to 8 USD).

The interview guide for PWTB (S1 File) included questions about the following: their perceptions of facilitators and barriers to using DATs and differentiated care; experience with using the DATs; stigma; disclosure; and recommendations for improving the DATs and differentiated care. The interview guide for HCWs and stakeholders (S2 File) asked questions on their roles in the implementation of DATs and differentiated care; motivators and barriers of implementing DATs; how DATs impacted their work; negative changes brought by DATs and how they can be addressed; perceptions on DAT training provided and suggestions for future training; willingness and feasibility to continue implementing DATs with differentiated care; who should implement DATs; sustainability; gaps in implementation and how it can be improved.

Interviews were conducted in the dominant languages spoken in the study areas (Setswana, isiZulu, Sesotho or English) based on each participant's preferred language. All interviews were transcribed verbatim and back translated to English for interviews conducted in local languages. Transcriptions and translations were done by SK and other trained research staff fluent in the study languages. SK, TD and FM checked the accuracy of the transcripts against digital recordings. Transcripts were not given to the participants for comment or correction.

## Ethical considerations

The study protocol was approved by the World Health Organisation Research Ethics Review Committee (0003296), University of the Witwatersrand Human Research Ethics Committee, Johannesburg (Reference 200102) and the London School of Hygiene & Tropical Medicine Research Ethics Committee (19135−1). All participants provided written informed consent to participate in the study and be recorded during the IDIs. Audio recordings, field notes and transcripts, were identified only by a unique study number. All study records were stored securely in locked filing cabinets where access to the records is restricted to specified study team members.

## Data analysis

A codebook was developed by three researchers TD, LD and FM using inductive coding. Data were analysed using NVIVO software. Thematic inductive analysis was used [19]. The process of refining, reviewing codes and emerging themes was repeatedly done until no additional themes could be identified [19,20]. We used the Unified Theory of Acceptance and use of Technology (UTAUT) framework to summarise the study findings [21,22]. The UTAUT identifies four

broader constructs that explain technology acceptance: performance expectancy, effort expectancy, social influences, and facilitating conditions. Performance expectancy refers to perceived usefulness, which is the degree to which an individual believes that the technology will help them. Effort expectancy, or ease of use, refers to how easy the technology is to use. Social influences refer to the influence that other individuals for example, family or community members have on someone's ability to accept or use the technology. Facilitating conditions refer to the quality of the organizational infrastructure that exists to support individuals using the technology [21]. Findings were supported by anonymised direct quotes extracted from interviews. The COREQ (Consolidate criteria for REporting Qualitative research) checklist was used to check the content of this manuscript (S3 File). Participants did not provide feedback on the findings.

## Findings

Thirty-seven PWTB and 35 HCWs and stakeholders were approached. In both groups, two people declined to participate, leaving 68 interviews in total with 35 PWTB, and 33 HCWs and stakeholders (Table 1). Reasons for non-participation

**Table 1. Participant's profile.**

|  | N(%) | Min-max |
|---|---|---|
| **People with TB** | **35** |  |
| *Sex* |  |  |
| Males | 20(57) |  |
| Females | 15(43) |  |
| *Adherence* |  |  |
| Non-adherent | 12(34) |  |
| Adherent | 20(57) |  |
| Opted out | 3(8) |  |
| *DAT used* |  |  |
| Box | 20(57) |  |
| Labels only | 11(31) |  |
| Switched from label to box | 4(11) |  |
| *Age, years* |  | 22-83 |
| **HCWs, treatment supporters and stakeholders** | **33** |  |
| *Sex* |  |  |
| Males | 6(18) |  |
| Females | 27(82) |  |
| *Job Category* |  |  |
| TB Nurses | 13(39) |  |
| Community Health Workers | 4(12) |  |
| General Worker | 1(3) |  |
| Treatment supporters | 2(6) |  |
| Interns | 6(18) |  |
| District Stakeholders | 6(18) |  |
| Provincial Stakeholders | 1(3) |  |
| *Age, years* |  | 27-59 |

Our findings were articulated under six major themes and 22 subthemes as described below. The themes were inductively derived from the data. Our study did not reveal any sex differences in acceptability of either labels or smart pillbox.

were being busy, not wanting to be recorded and wanting to know effective results before sharing their perceptions. In all interviews, none of the participants withdrew participation during the interview and each participant was interviewed once.

## Facilitating factors of smart pillbox and differentiated care among PWTB

**Favourable features of the box.** Participants who used the smart pillbox appreciated both the alarm reminder which enabled them to take medication every day at the same time, and the clinic visit reminder showed by the illuminating yellow light. Participants believed that their treatment adherence improved and would have favourable treatment outcomes because of the alarm reminder.

Furthermore, some PWTB reported that the smart pillbox was easy to use because it was portable, they appreciated the long battery life and that it could be used by someone without a cell phone. Not requiring a cell phone enabled the smart pillbox to be used by homeless PWTB, and HCWs reported that some of them completed treatment because they were supported by the DAT. Participants also appreciated the storage to keep their TB medication safe. Others with comorbidities reported that they stored other medication in the smart pillbox. These features made it possible for some participants to adapt it to their lifestyles, for example, some would be reminded by the smart pillbox to take out medication before going to work and then ingest medication at work later. The reported favourable features of DATs showed the performance expectancy (ease of use) of the DAT.

*...I loved storage and reminder to drink medication. Our medication stays safe in the box. You will find the sugar diabetes medication there too, you understand? (Female, 38 years, Smart pillbox, opted out).*

**Social support.** Some participants reported the importance of social influences as a facilitator to using DAT. Some PWTB reported that their families supported the idea of using the smart pillbox. Others would be alerted by family members when the smart pill box alarm went off if they were away from it. Some participants reported that they did not fear community stigma indicating that those who were surrounding them were supportive.

*The box helps me because of the alarm so when I oversleep my son wakes me up and say, "daddy, daddy, the box is ringing", then I open it and take my medication. Things like that. (Male Participant, 33 years, Smart pillbox).*

**Adequate knowledge of DAT and counselling.** Adequate knowledge of the DAT and counselling was reported as one of the facilitating conditions for DAT and differentiated care use. Some PWTB were satisfied with the explanation they were given during enrolment as well as the adherence counselling that they received.

## Barriers of using the smart pillbox and differentiated care among PWTB

***Unfavourable features of the smart pill box.*** Some participants who used the smart pillbox reported unfavourable features, such as poor portability when travelling. While PWTB using the smart pillbox chose the time they wanted the alarm to go off, some participants mentioned that the alarm sounded at inconvenient times for both themselves and their family members. These unfavourable features limited the effort expectancy of the DAT.

*Yes, it [referring to the box] would awaken people, and make noise when they are asleep (Female, 28 years, smart pillbox).*

**Malfunctioning box and SMS problems.** The effort expectancy of DAT for some participants was limited by malfunctioning of the DAT. Some participants noted malfunctions with the smart pillbox, experiencing issues such as, multiple alarms, alarms activating at incorrect times, or no alarm activation at all. These technical glitches were a source of irritation for many participants, and they caused some PWTB to withdraw from the study and return the smart pillbox.

The SMS reminders for a missed dose also malfunctioned at times, with PWTB receiving messages even after having taken medication using the smart pill box, causing further irritation among some participants.

*It [alarm] worked properly for 2 weeks then after two weeks it changed. Sometimes it would ring 30 minutes before or 30 minutes after. When it started losing time, I saw that it will mislead me, and I returned it [smart pill box]. (Male Participant, 56 years, Opted Out).*

However, it is worth noting that participants who withdrew from the study due to the malfunctioning of the smart pillbox still appreciated the importance of the smart pillbox when it functioned properly. One participant reported that they started taking medication at erratic times after withdrawing from the study but when they had the smart pillbox, they would consistently take it at the same time.

*When I had the box, I would take my medication on time 100 percent, but now I know at times I do not take it at the exact time because sometimes I'm late, when I check it's 10am and I should have taken my medication at 5am. (Female Participant, 38 years, opted out).*

**Perceived stigma.** Concerns about stigma prompted some participants to remove medication and leave the smart pillbox behind when traveling. PWTB were apprehensive that the DAT might inadvertently reveal their TB status due to alarms or inquiries about the purpose of the smart pillbox. While some PWTB valued home visits, there was a perception that certain individuals avoided them to prevent drawing attention from neighbours.

*That one [home visits] brings me unnecessary attention. Sometimes, my neighbor does not know I am taking treatment, and we know when the van from the clinic stops at your gate, we know the person didn't take her medication they will say "she doesn't drink her medications, they came for her" you see, that's home visits (Female Participant, 26 years, Smart pillbox).*

**Facilitating factors of the labels and differentiated care among PWTB**

***Favorable features of the labels.*** Participants who used the label felt appreciated when they received the acknowledgement SMSs.

*Using the stickers [labels] was very helpful because you get an SMS saying, "thank you for taking your medication". So, it's good; you feel great. (Male Participant, 46, Label).*

Younger participants felt the labels were easy to use and the instructions were easy to follow. Some PWTB reported that the label was portable and private because they were just using their phones and not carrying anything additional.

*The stickers [labels] are simple, especially for someone who is working. It is simple. To be honest, you will not go to work carrying the box. (Female Participant,43 Label).*

**Social support.** Participants reported that they received support from family, community, and co-workers. Some PWTB mentioned that their families supported the idea of using labels. When facing challenges in sending the SMS, some participants were assisted by family members and neighbours, and some of those who forgot to send the SMS were reminded by family members and co-workers.

*Yes, so suddenly you will hear one of my work colleagues saying to me "hey, did you take that nyaope (referring to TB medication) of yours, did you SMS now?" Oh, that's when I remember that oh, I drank my medication, but I didn't send [the message]. Then that's when I would send (Male Participant, 44 years, Label).*

**Barriers to using labels and differentiated care among PWTB**

*Individual factors.*  Effort expectancy (ease of use) of labels was hindered by some individual characteristics. Some PWTB using labels were not able to send the SMS due to limited familiarity with technology and illiteracy, and this was primarily observed among older participants. Some participants forgot to send the SMS, while others perceived the process of sending the code as unnecessarily burdensome. Others lacked understanding of the purpose of the labels, some did not find the label useful because it did not remind the participant before taking medication. The participant had to act first and then receive a confirmation SMS later.

*I would say there is no use to send the SMS because I drank my medication, you see? (Male, 38 years, Labels).*

**Access to mobile phones.**  Some participants were sharing phones with family members and would be unable to send the SMS if the phone was not with them. Others lost or damaged their phones which stopped them from sending the code after taking medication. Other participants ended up switching to using the box because of the mobile phone access challenges.

*I didn't use the stickers [labels] during that time because I had lost my phone, do you get me? I arrived there and told sister [TB Nurse], that's when they gave me the box (Male, 30 years, Switched from label to box).*

**Technical problems.**  For PWTB who were using labels, there were some codes which were not received by the platform due to technical glitches which were unresolvable. Also, some PWTB who used the labels were not able to send the code because they needed a positive airtime balance for the SMS to go through even though the airtime was not deducted. This led to participants receiving excessive reminder SMSs, which participants found irritating.

**Contextual challenges.**  Contextual challenges, such as poor network coverage in certain areas, which were further compounded by regular power cuts planned by the electricity provider (referred to as load shedding), preventing participants from charging their phone batteries, made it difficult for the participants to send the SMSs. These issues not only affected DATs but also disrupted differentiated care as implementing staff struggled to reach PWTB over the phone. The inability to send codes for various reasons led to PWTB receiving excessive reminder SMSs, causing irritation among some participants. In response to these challenges, some PWTB opted to switch to using the smart pillbox.

*When there is load shedding, the network is quite low and also sometimes the phone battery might be flat, so you are unable to send the code (Female, 22, Labels).*

**Technology fatigue.**  Other participants who used the labels experienced technology fatigue whereby after taking medication, they would intentionally choose not to send the SMS due to exhaustion from the overuse of technology. Consequently, some of these participants ended up switching to using the smart pillbox.

*I was lazy to SMS. I would take my medication but would not SMS because I was just tired. (Female, 51, Switched from label to box).*

*Positive perceptions of DATs and differentiated care among HCWs and key stakeholders*

*Ease of use.*  HCWs and implementing interns reported that the adherence platform was user-friendly. The HCWs reported that the box was easy to implement and echoed favourable features such as the alarm and storage. They also noted that some of the participants who received the smart pillbox were homeless, and the box assisted them to complete their medication.

**Perceived benefits of remote monitoring using DATs.** Real-time monitoring using DATs enabled HCWs to identify non-adherence early and intervene accordingly on time which improved efficiency in TB care.

*If you have a patient who's struggling with something, it will be easier for you to know whether the patient is defaulting or whether there's a problem with regards to the box that you gave, so you will know quickly, because if the patient doesn't open the box for a day then you will know something is up, and you have to go and phone the patient. They will tell you, "sister, I'm struggling with this thing. It's not that I'm not taking my medication." (Female TB Nurse, 35 years).*

Some HCWs appreciated that DAT saved time and resources to implement community-based DOT and decongested the clinics. One HCW explained how DAT helps in reducing the number of PWTB who are lost to follow up. Without DAT, such PWTB might discontinue their treatment and subsequently return to the facility sick, resulting in facilities spending more resources to test and initiate treatment again. The comparison between DAT and DOT came from routine HCWs and stakeholders because they had experience with both.

*Mmm. So, I am happy, and I just need these devices to stay just to help us not to have those loss to follow, you know, lost to follow is a problem. The patient will come back very sick, very ill. Now you start them from zero, you start giving them, you start initiating, and it's a lot of money again because you collect again sputum. It's like you starting a new patient. You collect sputum from scratch (Female TB Nurse, 49 years).*

HCWs felt that the use of DAT and differentiated care reduced their workload since TB nurses did not have to go through patient files to check information. Both PWTB and HCWs recommended use of DATs among PWTB and other diseases, citing potential benefits for their respective populations. HCWs expressed their willingness to continue implementing DATs as they found it beneficial.

*With the device, it's very easy. It's very quick and unlike going to the file or to that book of mine checking, it's a long process. (TB Nurse, Female, 49 years).*

A good client-health care worker relationship was reported to be a positive experience when using DAT. Some participants who received differentiated care such as reminder messages, phone calls and home visits felt supported by the HCWs.

**HCWs negative experiences with DAT and differentiated care implementation**

**Concern of PWTB using DAT but not ingesting medication.** Even though HCWs generally had positive experiences with the smart pill box, some reported the challenge of noticing that some PWTB were using the DAT but not taking medication which was a concern regarding the usefulness of DAT.

*The device showed that the patient was adhering well because she was opening the box every morning as if she's taking the treatment, but we heard from the family that actually she's not taking the treatment. Uhm… they said it as it is that they are staying with the patient and then the patient will open the box as the box alarms, but she doesn't take the treatment. (Female TB Nurse, 35 years).*

**Challenges with implementing DATs and differentiated care.** HCWs reported encountering difficulties in implementing labels, citing individual challenges such as the inability of PWTB, especially among the elderly and illiterate population, to send the SMS. Technological challenges including the requirement for a positive airtime balance, and contextual challenges like lack of electricity, further contributed to complicating the implementation of labels. These

challenges would manifest as instances of non-adherence on the adherence platform, necessitating HCWs to engage in persistent follow-ups with the participants.

HCWs also reported challenges with implementing DATs among drug users because some lose the box, and they move from home to staying on the streets which makes it difficult to follow up on them. In addition, it was reported that differentiated care was difficult to implement among highly mobile people because mobility made it difficult to conduct home visits. Frequent change of phone numbers was also a challenge.

**Perceived stigma among HCWs.** HCWs and stakeholders also perceived stigma as a barrier to DAT and differentiated care implementation.

*I don't want to lie, I emphasized to patients all the time that they must give correct address and contact details so that we are able to trace them, but they continue to provide wrong details. I think maybe they don't want to be traced or get home visits. Maybe it is the stigma around TB; maybe they are trying to hide their TB status to their loved ones. I'm thinking that could be the reason (Female TB Nurse, 31 years).*

HCWs perceived that one of the reasons for refusal to participate in the main trial was fear of stigma while using the smart pill box. Moreso, in terms of differentiated care, HCWs reported that they had a challenge with some participants providing wrong contact details (address and phone number) because they feared stigma, so they did not want to be phoned or visited.

**Necessary conditions to scale up DAT.** Stakeholders and HCWs generally felt they would want the implementation of DATs to continue. However, they shared some necessary conditions that must be in place for DATs to be implemented more smoothly. Implementing staff reported that they sometimes lacked human resources and transport to conduct home visits early. There is a need for financial and human resources to continue implementing the DAT and differentiated care. HCWs appreciated the support they had from interns assigned to them during the study and would want to continue having similar support when implementing DAT. Facilities could identify existing staff such as data capturers to assist with charging batteries and registering PWTB. Also, capacity building whereby all HCWs are trained and receive refresher training, is key. Stakeholders also reported the importance of integrating DAT with existing electronic TB registers.

*"I think there should be funds to sustain because this is very a beautiful project. For them to sustain it, it would mean that boxes or the containers must always be there, treatment must always be there, people who are trained to do that to continue with this project are always there, data capturers are always there. It can then be implemented and be linked to Tier and then there's a person who is always responsible for checking that". (Stakeholder 5, Female, 48 years).*

Community education is needed to combat stigma as well as comprehensive psychosocial support to holistically provide care. PWTB made some suggestions to improve the smart pill box, some wanted a smaller size, a different darker colour.

The barriers and facilitators of DATs usage we found from our interviews are summarized in Fig 1 using the UTAUT framework.

## Discussion

We identified barriers and facilitators influencing the acceptability of two DATs and differentiated care in two South African provinces. Our study demonstrated that DATs can be an important person-centered tool that can support TB treatment adherence. Acceptability of the smart pillbox was strongly facilitated by its ease of use across age groups. Features of the smart pillbox were generally found to be acceptable despite the challenges which can be addressed. This is consistent

**Barriers**
**(Low use and acceptance)**

**Facilitators**
**(High use and acceptance)**

Participants not understanding the purpose of sending SMS's* (label)

No reminder on label *

PWTB using DAT and not ingesting medication # (box and label)

**Performance expectancy**
**(Perceived usefulness)**

PWTB feeling motivated/appreciated by the confirmation message (label)*

Belief that DAT helps in improving adherence and treatment outcomes  (box and label).*#

Good patient-HCW relationship(box and label)*#

Perceived improved efficiency in TB care (DAT saves time and resources)#

PWTB reminded (box alarm, SMS, yellow light) *#

Preferred support action differs (box and label) *#

 Recommending DAT to other PWTB and other diseases(box) *#

Phone sharing /no access (label )*#

Old age, illiteracy (label) *#

Lack of portability of the box *#

Poor network/ power cuts (label)*#

Forgetting  to send SMS (label)*#

 Incorrect reminder SMS (box and label)*

Malfunctioning (box)*

Box alarm is noisy and irritating*

**Effort expectancy (perceived ease of use)**

Box easy to use for all ages *#

Labels easy to use for the young *

Portability of DAT (box and label) *#

Privacy (label) *

Storage (box) *#

No need for a phone (box) *#

Everwell platform easy to use #

DAT reduces workload #

Lack of social support*

Perceived stigma*#

**Social influences**

Social support when using DAT (family, friends, co-workers)*

Insufficient knowledge (label)*

**Facilitating factors**

Adequate knowledge and counselling*#

* Findings from PWTB only
# Findings from HCWs and stakeholders
*#Findings from HCWs, stakeholders and PWTB

**Fig 1. Key findings on facilitators and barriers to DATs acceptability based on the UTAUT framework.**

with findings from a study on the acceptability of the smart pillbox among drug resistant patients in South Africa [13]. Some concerns included lack of portability and the loudness of the alarm. The challenge of loud alarm volume has been reported in other studies in South Africa and India [14,22]. The finding that PWTB would leave the smart pillbox behind and carry medication, then ingest it later, is consistent with other studies from Vietnam and South Africa [14,23]. Future implementation should include providing different pillbox sizes to suit the needs of PWTB and the alarm volume which can be adjusted to the individual's preference. This will enhance the person- centeredness of the DAT.

Favourable features of the labels such as privacy, portability and ease of use among the young were the acceptable features of the labels. However, some PWTB faced individual, technical, and contextual obstacles which made it difficult to implement, hence some PWTB switched to using the smart pillbox. Barriers such as lack of access to cell phones, power cuts and poor network were also reported in India and Uganda [24,25]. In South Africa, the challenge of PWTB needing positive airtime balance and codes not going through, could not be resolved during the trial and that was a major challenge to implementation. We recommend that if this method is to be used again, it is important to engage all telecommunications service providers to ensure that the service does not need a positive airtime balance. The reported challenges on the implementation of the labels show that even if the technical glitches are resolved and the technology is working perfectly, there will be many PWTB who will not be able to effectively use the labels due to factors such as old age, illiteracy, lack of phone access, lack of electricity, etc. This suggests that labels should not be implemented on their own but offered as an option in addition to the pillbox. This will provide an option to those who are uncomfortable with always carrying the smart pillbox and the tech-savvy younger PWTB. Strengthening education on the importance of sending the SMS is important.

Stakeholders and HCWs expressed willingness to continue using the DAT due to the positive attributes which included remote monitoring (thought to save time and reduce workload) and the possibility of early intervention (instead of waiting for a missed clinic visit appointment to act). Ease of use of the adherence platform was also reported by HCWs in Uganda and India [26,22]. Financial and human resources will be needed to support implementation. Even though TB nurses felt DATs reduce workload, they also reported they would still need additional support to implement DATs. Participants recommended that facilities could identify existing staff such as data capturers to assist with charging batteries and registering PWTB. We also found that integrating DAT with existing systems such as Tier.net (Three Interlinked Electronic Register for TB & HIV) will make TB management more efficient. Capacity building is important, and this can possibly be done at subdistrict level. If subdistrict coordinators are trained, they will be able to train facility staff and staff rotation will not affect implementation much because the subdistrict coordinator will be there to train and support new TB staff.

Differentiated care was appreciated by PWTB and HCWs, however some found the home visits to be stigmatizing. The finding that participants preferred different support actions strengthened the need for differentiated care.

Fear of stigma poses a challenge to TB Care in terms of DAT use and disclosure. This finding is similar to the findings of studies conducted in South Africa, India and Uganda [27,14,22]. This raises the issue of increasing anonymity when conducting any home visits for TB, which can be done by not wearing uniforms and not using branded vehicles, when possible.

We would recommend increased TB awareness in the communities to reduce stigma. This would also encourage PWTB to disclose their TB status to family and friends and receive support. We did not find any differences in acceptability between genders. This is similar to two multi-country studies which found that gender did not impact the perceived acceptability or feasibility of the 99DOTS and smart pillbox [28,29]. However, a study in Uganda showed gender differences in acceptability with women with TB reporting to have faced more barriers to engaging in 99DOTS than men with TB, mostly due to more limited access to a cellphone and feared accidental disclosure of their TB status and abandonment [24]

## Strengths and limitations

One of the study strengths was the large sample of PWTB, HCWs, implementing research staff and stakeholders from different provinces in South Africa so we ascertained perceptions from diverse groups. For example, routine HCWs were

able to compare DOT and DATs because they experienced both whereas research staff shared their perceptions on DAT only. Secondly, we interviewed adherent, non-adherent and PWTB who withdrew from the main study to understand if their perceptions on DAT and differentiated care differed. The study was not without limitations. DATs acceptability may have been over reported because of social desirability bias from both PWTB and HCWs. The existing relationship between the HCWs and interviewers might have increased social desirability. The interviewers built rapport with participants during the informed consent process to reduce social desirability bias.

## Conclusion

Our study provided evidence on facilitators and barriers of DATs from the perspective of PWTB, HCWs and stakeholders in the South African context. Our study showed that the use of DATs and differentiated care generally improves person-centeredness in TB care. Some of the barriers and facilitators were DAT specific. Even though there were factors inhibiting the use of the box such as portability and malfunctioning alarm, some participants found features such as audible alarm reminder and storage acceptable. Some features unique to the labels such as privacy and portability facilitated their use. However, implementing labels was a challenge for both PWTB and HCWs as users of the labels experienced several challenges that made some of them switch to using boxes. Our study showed the importance of differentiated care and asking PWTB their preferred way of being contacted. Further work is needed to make DATs and some of the differentiated care components less stigmatizing. We recommend that policymakers should consider acceptability, differing situations, and allowing flexibility to possibly increase uptake and utilization of DATs.

## Supporting information

**S1 File. Interview guide for PWTB.**
(DOCX)

**S2 File. HCWs and stakeholders.**
(DOCX)

**S3 File. COREQ (COnsolidated criteria for REporting Qualitative research) checklist.**
(PDF)

**S4 File. Transcripts.**
(ZIP)

## Acknowledgements

We would like to thank the following: Tshwane and Bojanala districts for allowing us to conduct the study in their districts. Research assistants and interns who conducted the interviews namely Percy Phalane, Nokukhanya Shabalala, Kgomotso Malatji, and Letta Raseroka, Lesley Blennis and Dumazile Xaba. Youth Health Africa interns who implemented the study. All the transcribers who dedicated time to capture the conversations verbatim. All study participants who consented to participate in this study.

## Author contributions

**Conceptualization:** Tanyaradzwa Nicolette Dube, Katherine Fielding, Degu Jerene, Salome Charalambous.

**Data curation:** Tanyaradzwa Nicolette Dube.

**Formal analysis:** Tanyaradzwa Nicolette Dube, Fezeka Mboniswa, Liza de Groot, Siyanda Khumalo.

**Funding acquisition:** Salome Charalambous.

**Investigation:** Noriah Maraba, Katherine Fielding, Degu Jerene, Salome Charalambous.

**Methodology:** Tanyaradzwa Nicolette Dube, Katherine Fielding, Degu Jerene, Salome Charalambous.

**Project administration:** Fezeka Mboniswa, Nontobeko Ndlovu, Nontobeko Mokone, Lebohang Masia.

**Resources:** Nontobeko Ndlovu, Katherine Fielding, Degu Jerene, Salome Charalambous.

**Software:** Salome Charalambous.

**Supervision:** Nontobeko Ndlovu, Degu Jerene, Salome Charalambous.

**Validation:** Fezeka Mboniswa, Liza de Groot.

**Writing – original draft:** Tanyaradzwa Nicolette Dube.

**Writing – review & editing:** Tanyaradzwa Nicolette Dube, Liza de Groot, Noriah Maraba, Katherine Fielding, Degu Jerene, Zewdneh Shewamene, Bianca Gonçalves Tasca, Adrian Leung, Salome Charalambous.

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
