## [Decision Letter · Decision Letter 0]

18 Feb 2025

PONE-D-24-05025Acceptability of Digital Adherence Technologies to support people with drug-susceptible TB in South AfricaPLOS ONE

Dear Dr. Dube,

Thank you for submitting your manuscript to PLOS ONE. After careful consideration, we feel that it has merit but does not fully meet PLOS ONE’s publication criteria as it currently stands. Therefore, we invite you to submit a revised version of the manuscript that addresses the points raised during the review process.

**ACADEMIC EDITOR: ** This manuscript provides a timely and important perspective on the importance of differentiated TB care  and the potential use of DATs to support adherence in a South African context. The analysis is detailed and practical in nature. The authors are to be congratulated. There are a few minor revisions or points of clarification, which will further strengthen your submission. However, overall, this is a strong piece of research. In addition to reviewer suggestions, which you will find below, please make the following recommended revisions: 

In the abstract, write 'and/or' instead of 'and or' (line 28)Proofread spacing around references (see line 110 for example) and also remove full-stops for titles and sub-headingsElaborate more on what was done to manage the pre-existing relationships between data collectors and interviewees mentioned in lines134-5.  Did data collectors reflect on this in field notes? Was this discussed during the consent process. Consider revisiting this in the limitations section as well.The idea of sampling to reach data saturation was mentioned in line 140. However, this was not addressed in terms of how the final sample was decided. Clarify if data analysis was happening concurrently with data collection so that sampling ended once saturation was met or if other factors influenced the final sample. The term saturation in the analysis section (line 166) seemed to imply a different concept of saturation, involving the three coders rather than something that informed sampling. Clarify the language(s) of interviews and who was involved in the translation and transcription process (a service or authors) in lines 152-53Specify a reference for the qualitative analysis (if grounded in a particular form of thematic analysis) on line 165Move up the UTAUT reference to line 168.

We look forward to receiving your revised manuscript.

Kind regards,

Sara Jewett Nieuwoudt, Ph.D, MPH

Academic Editor

PLOS ONE

Journal Requirements:

2. In the online submission form, you indicated that the data analyzed during for this manuscript is available from the corresponding author upon reasonable request. Our complete transcripts contain data that is sensitive or includes identifying information. We would like the confidentiality of the participants protected in accordance with the consent agreement. Due to these

concerns, we are unable to make the full transcripts available to a wider audience. We will make the transcripts available to fellow researchers or reviewers who complete a data sharing agreement.

Reviewers' comments:

Reviewer's Responses to Questions

**Comments to the Author**

1. Is the manuscript technically sound, and do the data support the conclusions?

Reviewer #1: Yes

Reviewer #2: Yes

Reviewer #3: Yes

2. Has the statistical analysis been performed appropriately and rigorously? 

Reviewer #1: N/A

Reviewer #2: Yes

Reviewer #3: Yes

3. Have the authors made all data underlying the findings in their manuscript fully available?

Reviewer #1: Yes

Reviewer #2: Yes

Reviewer #3: No

4. Is the manuscript presented in an intelligible fashion and written in standard English?

Reviewer #1: Yes

Reviewer #2: Yes

Reviewer #3: Yes

5. Review Comments to the Author

Reviewer #1: Overall, this is a well written article which demonstrates the potential for these technologies to assist with TB medication adherence in South Africa. Minor comments are below:

Abstract

-Recommendation to add a sentence or two about the number of people with TB globally and/or in South Africa.

Intro

-You need a sentence or two to say something about TB in South Africa (see below for line 110).

-line 60: spell out health care workers

-Briefly say what DOT is

Methods

-line 110” “In 2022 the TB incidence in South Africa was 280K.” This belongs in the introduction.

-line 111: the districts were Tshwane and Bojanala? I would just say that.

-line 111: “conducted at public health clinics providing care”. I would change to something like “conducted at public health clinics where care for TB was offered…” The former sounds like the clinics themselves were providing care.

-Data collection: what is the definition of “good adherence” vs “poorer adherence”. This needs to be stated somewhere in methods.

-Data collection: line 114 you say intended to conduct 20 interviews which makes one think that you did not do this many interviews? If you did do this number of interviews, be direct and just say we conducted x interviews. If not, you can say you intended to do so many, but only ended up doing x because of whatever reason. It’s also confusing that in the findings section the numbers are different as you say 37 PWTB and 35 HCW were approached. Clarification would be helpful.

Findings

-Line 180: 37 PWTB, 35 HCW + stakeholders were approached. Be consistent. If you are going to split out HCW and stakeholders in other sections of the paper please do that throughout. Table 1 also lumps HCW & stakeholders together. You need to decide if you want to treat these groups as separate or lump them together.

-Line 180: The issue above also makes it difficult to follow the math for the number of participants. 2 people declined from each group so 37-2=35 and 35-2=33 but then somehow we end up with 26 HCWs which does not add up.

Reviewer #2: It is good to dig into approaches increase patients adherence to long treatments for killer diseases like Tuberculosis. However these technologies were or how much successful they were, it is paramount to share experiences so other researchers can develop more or find alternative approaches.

Reviewer #3: This is a qualitative study from South Africa in which the authors evaluated the acceptability of digital adherence technologies or DATs to support people receiving treatment for drug-susceptible tuberculosis (PWTB). There have been few studies examining the acceptability and feasibility of DATs among PWTB, HCWs and stakeholders in low and middle-income countries. Therefore, the authors set out to investigate the key factors associated with the use of smart pill boxes and labels alongside differentiated care in adults receiving treatment for drug-sensitive TB, in healthcare workers, and stakeholders in South Africa.

The study was nested within the Adherence Support Coalition to End TB (ASCENT) trial, a multi-country cluster randomized trial in which two DATs (a smart pillbox and a cellphone-based strategy) combined with differentiated care were used to support PWTB. The present study’s focus is on South Africa. In-depth interviews were conducted with purposively selected PWTB, healthcare workers, and stakeholders. Data were analyzed by the inductive thematic analysis and Unified Theory of Acceptance and Use of Technology was the framework used to synthesize and summarize the findings.

The authors main findings were that facilitating factors for the pillbox were alarm reminder, storage, ease of use, not requiring a phone, social support, and portability, whereas the barriers were its lack of portability for some of the PWTB and the fact that it malfunctioned. As to the use of labels with SMS, facilitating factors were ease of use among the young, daily confirmation messages, social support, privacy, and portability. Barriers to the use of labels were older age, illiteracy, forgetting to send the SMS, lacking understanding, no cell phone access, power cuts, and unresolvable technical glitches. The authors highlight that the use of DATs and differentiated care may be stigmatizing and that further work may be needed to make them less stigmatizing.

The manuscript is well written and provides a wealth of information on the methodology used for data analysis. The results are clearly presented and easy to interpret. Reasons for non-participation are described, and the participants profile are presented. Direct quotes are provided throughout the results that illustrate the themes well and support the authors’ interpretation.

I highly recommend this manuscript for publication as it is technically sound and provides a valuable understanding of factors that influence acceptability of DATs in PWTB in low- and middle-income settings.

6. PLOS authors have the option to publish the peer review history of their article (what does this mean? ). If published, this will include your full peer review and any attached files.

**Do you want your identity to be public for this peer review?** For information about this choice, including consent withdrawal, please see our Privacy Policy .

Reviewer #1: No

Reviewer #2: **Yes: ** LAYTH AL-SALIHI

Reviewer #3: No

---

## [Author Response · Author response to Decision Letter 1]

5 Aug 2025

Responses regarding data availability

Comment

Response

The data is available within the manuscript presented as quotes and deidentified transcripts are uploaded in supplementary files.

Comment

We note that this data set consists of interview transcripts. Can you please confirm that all participants gave consent for interview transcript to be published

Comment.

Yes, the participants consented to having their de identified data shared in published journals.

Responses to the Editors Comments

Response

Comment

Revision

Transcripts will be provided as supplementary information.

Comment

In the abstract, write 'and/or' instead of 'and or' (line 28)

Response

Done (line 29)

Comment

Proofread spacing around references (see line 110 for example) and also remove full-stops for titles and sub-headings

Response

Full stops removed on subheadings lines

Comment

Elaborate more on what was done to manage the pre-existing relationships between data collectors and interviewees mentioned in lines134-5. Did data collectors reflect on this in field notes? Was this discussed during the consent process. Consider revisiting this in the limitations section as well.

Response

To manage the relationship, the interviewers encouraged the HCWs to feel free to share their honest opinions and assured them that their responses were confidential. During interviews, the interviewers emphasized neutrality and created a respectful, non-judgmental space to mitigate social desirability bias and support candid engagement. (line 143-146)

Added the following in the Limitations:

The existing relationship between the HCWs and interviewers might have increased social desirability. (line 502-505)

Comment

The idea of sampling to reach data saturation was mentioned in line 140. However, this was not addressed in terms of how the final sample was decided. Clarify if data analysis was happening concurrently with data collection so that sampling ended once saturation was met or if other factors influenced the final sample. The term saturation in the analysis section (line 166) seemed to imply a different concept of saturation, involving the three coders rather than something that informed sampling.

Response

Saturation during data collection was achieved through regular debriefing discussions with the data collectors and interviews were stopped when no new information emerged. (line 150-153)

Comment

Clarify the language(s) of interviews and who was involved in the translation and transcription process (a service or authors) in lines 152-53

Response

Interviews were conducted in the dominant languages spoken in the study areas the participants preferred (Setswana, isiZulu, Sesotho or English) based on each participant’s preferred language. line (163-164)

Transcriptions and translations were done by SK and other trained research staff fluent in the study languages. SK, TD and FM checked the accuracy of the transcripts against digital recordings. line (166-168)

Comment

Specify a reference for the qualitative analysis (if grounded in a particular form of thematic analysis) on line 165

Response

Thematic analysis referenced (line 179)

21. Braun V, Clarke V. Using thematic analysis in psychology. Qual Res Psychol. 2006;3(2):77–101.

22. Saunders CH, Sierpe A, Von Plessen C, Kennedy AM, Leviton LC, Bernstein SL, et al. Practical thematic analysis: a guide for multidisciplinary health services research teams engaging in qualitative analysis. BMJ. 2023;

Comment

Move up the UTAUT reference to line 168.

Response

Reference moved to line 182

Responses to Reviewer 1

Comment

Abstract

Recommendation to add a sentence or two about the number of people with TB globally and/or in South Africa.

Response

Done

TB cases in South Africa added (line 23)

Comments

Intro

-You need a sentence or two to say something about TB in South Africa (see below for line 110).

Response

TB incidence reported (line 57-59)

Comment

Healthcare workers spelt out (line 61-62)

Response

-line 60: spell out health care workers

Comment

-Briefly say what DOT is

DOT is a monitoring strategy which can be facility-based, whereby patients visit a health facility daily to have a HCW observe their medication intake; or community-based, where a HCW or trained treatment supporter supervises the patient's treatment at home

(line 63-67)

Reference

Maher D, Mikulencak M. What is DOTS? What is DOTS? A Guide to Understanding the WHO-recommended TB Control Strategy Known as DOTS.

Response

Methods

Comment

-line 110” “In 2022 the TB incidence in South Africa was 280K.” This belongs in the introduction.

Response

TB incidence moved to introduction (line 58-59)

Comment

-line 111: the districts were Tshwane and Bojanala? I would just say that.

Response

Tshwane and Bojanala districts mentioned (line 98)

Comment

-line 111: “conducted at public health clinics providing care”. I would change to something like “conducted at public health clinics where care for TB was offered…” The former sounds like the clinics themselves were providing care.

Response

Changed to “conducted at public health clinics where care for TB was offered…” as suggested (line 118)

Comment

-Data collection: what is the definition of “good adherence” vs “poorer adherence”. This needs to be stated somewhere in methods.

Response

Good adherence was defined as 90% or more adherence and bad adherence was less than 90% adherence. (line 125-127).

Comment

-Data collection: line 114 you say intended to conduct 20 interviews which makes one think that you did not do this many interviews? If you did do this number of interviews, be direct and just say we conducted x interviews. If not, you can say you intended to do so many, but only ended up doing x because of whatever reason. It’s also confusing that in the findings section the numbers are different as you say 37 PWTB and 35 HCW were approached. Clarification would be helpful.

Response

Deleted the confusing numbers. We now have only mentioned the number approached and the number we finally interviewed under findings. (line 194)

Comment

Findings

-Line 180: 37 PWTB, 35 HCW + stakeholders were approached. Be consistent. If you are going to split out HCW and stakeholders in other sections of the paper please do that throughout. Table 1 also lumps HCW & stakeholders together. You need to decide if you want to treat these groups as separate or lump them together.

-Line 180: The issue above also makes it difficult to follow the math for the number of participants. 2 people declined from each group so 37-2=35 and 35-2=33 but then somehow, we end up with 26 HCWs which does not add up.

Responses

HCWs and stakeholders combined (line 128)

Agreed. I had separated HCWs from stakeholders (26 HCWs and 7 stakeholders=33). I have now merged the two to make it easier to read. (line194-196)

Response to Reviewer 2

Comment

It is good to dig into approaches increase patient’s adherence to long treatments for killer diseases like Tuberculosis. However, these technologies were or how much successful they were, it is paramount to share experiences so other researchers can develop more or find alternative approaches.

Response

Cluster randomized trials conducted in South Africa and China found improvement in adherence among PWTB who used the smart pill box (line 77-79)

---

## [Editor Report · Decision Letter 1]

27 Aug 2025

Acceptability of Digital Adherence Technologies to support people with drug-susceptible TB in South Africa

PONE-D-24-05025R1

Dear Dr. Dube,

We’re pleased to inform you that your manuscript has been judged scientifically suitable for publication and will be formally accepted for publication once it meets all outstanding technical requirements.

Kind regards,

Sara Jewett Nieuwoudt, Ph.D, MPH

Academic Editor

PLOS ONE

---

## [Editor Report · Acceptance letter]

PONE-D-24-05025R1

PLOS ONE

Dear Dr. Dube,

I'm pleased to inform you that your manuscript has been deemed suitable for publication in PLOS ONE. Congratulations! Your manuscript is now being handed over to our production team.

Kind regards,

on behalf of

Dr. Sara Jewett Nieuwoudt

Academic Editor

PLOS ONE